# Real time structural search of the Protein Data Bank

**Dmytro Guzenko[1], Stephen K. Burley[1,2,3,4], Jose M. Duarte[1]***

**1** RCSB Protein Data Bank, San Diego Supercomputer Center, University of California, San Diego, La Jolla, California, United States of America, **2** RCSB Protein Data Bank, Institute for Quantitative Biomedicine Rutgers, The State University of New Jersey, Piscataway, New Jersey, United States of America, **3** Cancer Institute of New Jersey, Rutgers The State University of New Jersey, New Brunswick, New Jersey, United States of America, **4** Skaggs School of Pharmacy and Pharmaceutical Sciences, University of California, San Diego, La Jolla, California, United States of America

* jose.duarte@rcsb.org

**Data Availability Statement:** Datasets used for training and benchmarking are available at https://github.com/rcsb/biozernike-validation.

**Funding:** The RCSB PDB is jointly funded by the National Science Foundation (DBI-1832184), the National Institutes of Health (R01GM133198), and

## Abstract

Detection of protein structure similarity is a central challenge in structural bioinformatics. Comparisons are usually performed at the polypeptide chain level, however the functional form of a protein within the cell is often an oligomer. This fact, together with recent growth of oligomeric structures in the Protein Data Bank (PDB), demands more efficient approaches to oligomeric assembly alignment/retrieval. Traditional methods use atom level information, which can be complicated by the presence of topological permutations within a polypeptide chain and/or subunit rearrangements. These challenges can be overcome by comparing electron density volumes directly. But, brute force alignment of 3D data is a compute intensive search problem. We developed a 3D Zernike moment normalization procedure to orient electron density volumes and assess similarity with unprecedented speed. Similarity searching with this approach enables real-time retrieval of proteins/protein assemblies resembling a target, from PDB or user input, together with resulting alignments (http://shape.rcsb.org).

## Author summary

Protein structures possess wildly varied shapes, but patterns at different levels are frequently reused by nature. Finding and classifying these similarities is fundamental to understand evolution. Given the continued growth in the number of known protein structures in the Protein Data Bank, the task of comparing them to find the common patterns is becoming increasingly complicated. This is especially true when considering complete protein assemblies with several polypeptide chains, where the large sizes further complicate the issue. Here we present a novel method that can detect similarity between protein shapes and that works equally fast for any size of proteins or assemblies. The method looks at proteins as volumes of density distribution, departing from what is more usual in the field: similarity assessment based on atomic coordinates and chain connectivity. A volumetric function is amenable to be decomposed with a mathematical tool known as 3D Zernike polynomials, resulting in a compact description as vectors of Zernike moments.

the United States Department of Energy (DE-SC0019749), grant recipient SKB. We gratefully acknowledge contributions from members of the RCSB PDB and our Worldwide PDB partners. The funders had no role in study design, data collection and analysis, decision to publish, or preparation of the manuscript.

**Competing interests:** The authors have declared that no competing interests exist.

The tool was introduced in the 1990s, when it was suggested that the moments could be normalized to be invariant to rotations without losing information. Here we demonstrate that in fact this normalization is possible and that it offers a much more accurate method for assessing similarity between shapes, when compared to previous attempts.

## Introduction

Structure similarity searching within the growing PDB archive [1, 2] revolutionized our understanding of protein evolution [3, 4, 5]. Over billions of years organisms in the natural world have generated stable, functionally useful three-dimensional protein shapes, which have been repeatedly reused on scales ranging from short structural motifs to oligomeric complexes. With more than 150,000 publicly-available PDB structures, efficient methods for detecting and quantifying protein structure similarity are essential.

Structure superposition tools were initially developed in the 1970s [6, 7] and the first algorithms for general structural alignment came in the 1990s [8, 9, 10], with more advanced methods appearing over the following decade [11, 12, 13, 14]. As the PDB grew, efficient searching of the entire archive became both important and difficult. Archive wide retrieval was first addressed by the Dali server [15] and subsequently by PDBeFold [16] and TopSearch [17] (see Hasegawa and Holm [18] for a review of the field). With a few exceptions, these methods have focused on the task of aligning single polypeptide chains or parts thereof.

Protein functional units are, however, not necessarily confined to the boundaries of domains or individual chains. They are often oligomeric, sometimes with multiple distinct quaternary structures resulting in similar functional units. Today, approximately half of the structures in the PDB are oligomeric (as of April 2020). In the wake of the 3DEM "resolution revolution" the fraction of oligomeric structures represented in the archive is growing year-on-year.

The ever-increasing amount of structural data, combined with rising complexity of the structures, requires development of faster, more accurate methods to process and classify structure similarity. Traditional comparison methods use atom level information, which can be complicated by the presence of topological permutations within a polypeptide chain and/or subunit rearrangement(s) within an oligomeric assembly. While solutions that address these problems exist [14, 19, 20, 21], they are computationally expensive and will not necessarily scale with continued growth of the PDB.

Alternative approaches looking beyond purely atomic information have been explored. One utilizes geometric descriptors, *e.g.*, interatomic distance distributions, yielding fast but less precise methods [22, 23]. Other related methods compare surface shapes. This research community has coalesced around the SHREC 3D shape retrieval contest [24], which occasionally features a protein track (most recently in [25]). For surface descriptor-based protein structure analysis, 3D-Surfer [26] has implemented a fast shape comparison service with numerous applications [27]. Surface descriptions of proteins, however, completely disregard information contained in the density distribution beneath the surface. These challenges can be circumvented by aligning and comparing electron density volumes. However, this has proven to result in a computationally-intensive search problem [28, 29, 30].

Herein, we exploit a 3D Zernike moment normalization procedure to implicitly orient electron density volumes and assess similarity in moment space with unprecedented speed. The general approach was suggested in [31] but has not been applied to date. Our normalization procedure produces rotation-invariant features that retain information about the shape of the

original object. Differences in features can be readily visualized. This approach to shape retrieval is highly performant, yielding structure alignments as byproducts of the normalization procedure. Since the method uses electron density volumes, it is agnostic with respect to topological differences in either the tertiary or quaternary structures.

Based on these principles, we have developed a search system that enables real-time retrieval of similar protein assemblies to a target assembly, obtained from the PDB or uploaded by a user, together with their alignment (http://shape.rcsb.org). The system uses coarse grained volumes created out of atomic models, though the method presented here is applicable to any kind of volume, be it experimental or simulated. An exhaustive search of 600,000+ bioassemblies and chains in the PDB requires less than one second on a single core of a typical CPU (*e.g.*, Intel Core i7-7567U), without precomputed clustering or results caching. The BioZernike software library used for normalizations and alignments is open source and freely available (https://github.com/biocryst/biozernike).

## Methods

We follow the derivations by Canterakis in [31], who introduced 3D Zernike polynomial decomposition of an arbitrary volumetric function $f(\boldsymbol{x})$ defined in the unit sphere (illustrated in Fig 1):

$$f(\boldsymbol{x}) = \sum_{n,l,m} \Omega_{nl}^{m} Z_{nl}^{m}(\boldsymbol{x}),\tag{1}$$

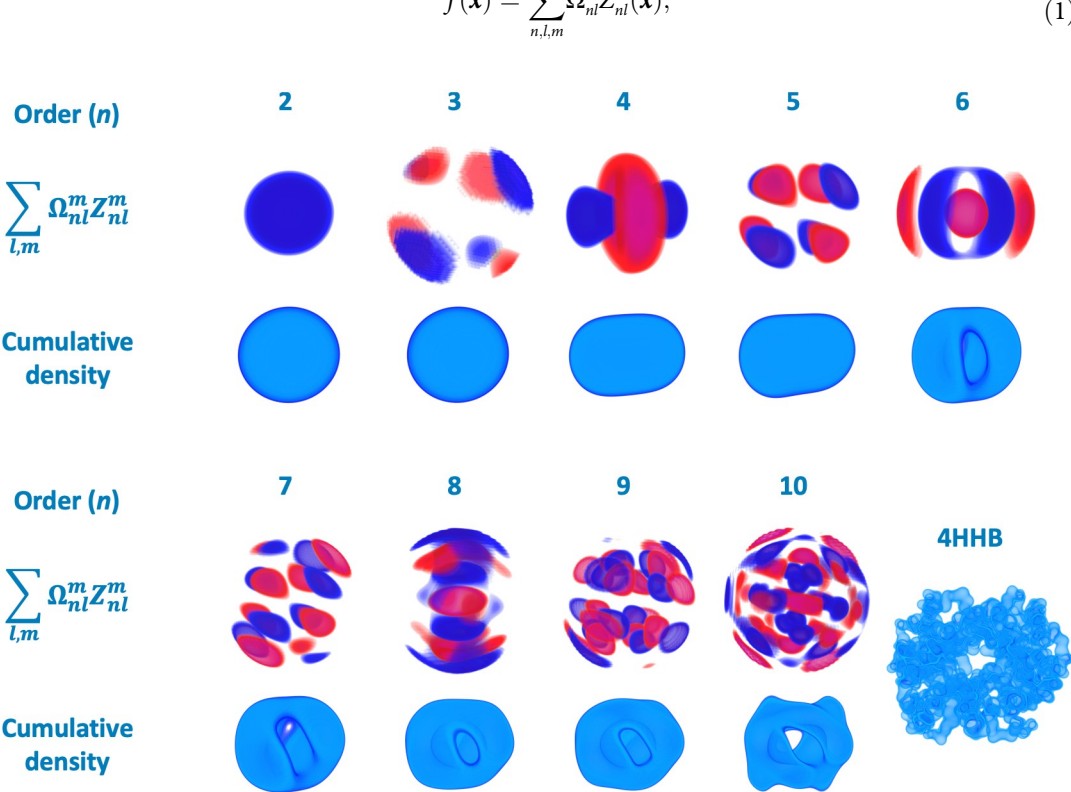

**Fig 1. Electron density decomposition into 3D Zernike moments, using human deoxyhaemoglobin (PDB ID 4HHB).** Top Layer: values of weighted 3D Zernike functions of order *n* in gradient from red (negative) to blue (positive). Bottom Layer: reconstruction of order *n* obtained by summation of all values up to order *n* (only positive density is shown).

where

$$Z_{nl}^m(\boldsymbol{x}) = R_{nl}(|\boldsymbol{x}|)Y_l^m\left(\frac{\boldsymbol{x}}{|\boldsymbol{x}|}\right) \tag{2}$$

are the (orthonormal in unit sphere) 3D Zernike polynomial functions, $\Omega_{nl}^m$ the corresponding moments, $n \in [0, N]$ (with $N$ the maximum polynomial order of the decomposition). $l$ and $m$ are the degree and order of the spherical harmonic functions $Y_l^m$, $l \in [0, n]$ so that $(n - l)$ is even, and $m \in [-l, l]$. $R_{nl}$ are the radius-dependent normalizing factors.

The moments can be expressed as a $(2l + 1)$-dimensional vector

$$\boldsymbol{\Omega}_{nl} = (\Omega_{nl}^l, \Omega_{nl}^{l-1}, \Omega_{nl}^{l-2}, \dots, \Omega_{nl}^{-l}) \tag{3}$$

whose norm corresponds to the trivial rotational invariant descriptor (3DZD) popularised by Novotni and Klein [32]:

$$F_{nl} = \parallel \boldsymbol{\Omega}_{nl} \parallel \tag{4}$$

## Canterakis norms for complete 3D zernike moment invariants

As in [33], we use $\zeta$-coding for 3D Zernike moment rotation. Briefly, given Cayley-Klein parameters $a$ and $b$ which define a rotation $R(a, b)$:

$$R(a, b) = \begin{pmatrix} \mathfrak{R}\{a^2 + b^2\} & -\mathfrak{I}\{a^2 - b^2\} & 2\mathfrak{I}\{ab\} \\ \mathfrak{I}\{a^2 + b^2\} & \mathfrak{R}\{a^2 - b^2\} & -2\mathfrak{R}\{ab\} \\ 2\mathfrak{I}\{ab^*\} & 2\mathfrak{R}\{ab^*\} & aa^* - bb^* \end{pmatrix}, \tag{5}$$

where $a, b \in \mathbb{C}$ and

$$aa^* + bb^* = 1, \tag{6}$$

and the *modified* 3D Zernike moments

$$\hat{\Omega}_{nl}^m = \frac{1}{c_l^m}\Omega_{nl}^m, \tag{7}$$

where

$$c_l^m = \frac{\sqrt{(2l + 1)(l + m)!(l - m)!}}{l!}, \tag{8}$$

the rotation can be expressed as:

$$(\hat{\boldsymbol{\Omega}}_R)_{nl}(\zeta) = \left(\frac{(a^*\zeta - b)(b^*\zeta + a)}{\zeta}\right)^l \cdot \hat{\Omega}_{nl}\left(\frac{a^*\zeta - b}{b^*\zeta + a}\right) \tag{9}$$

To obtain the rotated value for a particular $(\hat{\boldsymbol{\Omega}}_R)_{nl}^m$, we expand expression 9 and collect coefficients for $\zeta^m$. For example,

$$(\hat{\boldsymbol{\Omega}}_R)_{22}^2 = a^4\hat{\Omega}_{22}^2 - a^3b^*\hat{\Omega}_{22}^1 + a^2(b^*)^2\hat{\Omega}_{22}^0 + a(b^*)^3(\hat{\Omega}_{22}^1)^* + (b^*)^4(\hat{\Omega}_{22}^2)^* \tag{10}$$

As the SO(3) group has three degrees of freedom and Zernike moments are complex numbers, we follow by setting one moment and the imaginary part of another to zeros (see Fig 2a). In terms of computation, this corresponds to solving a system of two polynomial equations.

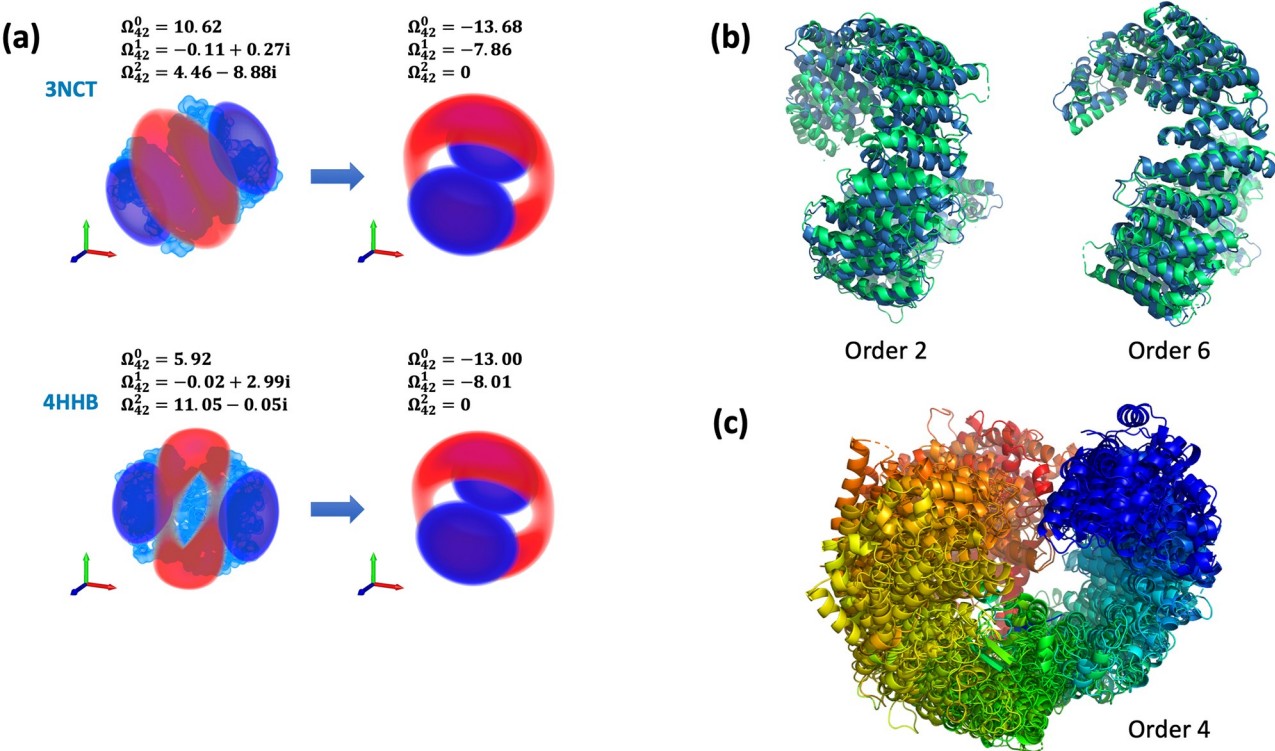

**Fig 2. Moment normalization.** (a) Rotational degrees of freedom are fixed by constraining values of chosen moments with respect to the 3D rotation group. The solution defines a rotation of the weighted 3D Zernike functions to a 'standard' position. (b) Alignment of two structures of human transportin 3: in unliganded form (4C0P) and in complex with ASF/SF2 (4C0O). Normalization order 2 is equivalent to alignment of the densities' principal axes. Normalization order 6 matches finer detail in the density, such as $\alpha$-helical bundles, at the expense of reduced overlap at the termini. (c) Multiple density alignment of 10 homologs of human transportin 3 as performed by the BioZernike library. Normalization order 4 is selected automatically with the alignment descriptor (see Methods).

Let us continue with the example 10 and fix the first two degrees of rotational freedom by setting $(\hat{\Omega}_R)^2_{22} = 0$. After substitution $\frac{a}{b^*} \rightarrow t$ we obtain a $4^{th}$ degree polynomial equation in $t$:

$$\hat{\Omega}^2_{22}t^4 - \hat{\Omega}^1_{22}t^3 + \hat{\Omega}^0_{22}t^2 + (\hat{\Omega}^1_{22})^*t + (\hat{\Omega}^2_{22})^* = 0 \tag{11}$$

that can be solved trivially using libraries of mathematical routines. (N.B. Both the coefficients of the polynomial and its roots are complex numbers).

Next, we fix one more degree of freedom by setting the imaginary part of another moment to 0. Let us choose $\Im\{\hat{\Omega}^1_{31}\} = 0$ for this example. Following algebraic manipulations, we obtain a $2^{nd}$ degree equation in $\frac{\Im\{b\}}{\Re\{b\}} \rightarrow s$:

$$(\Im\{t\}(\hat{\Omega}^0_{31} - 2\Re\{\hat{\Omega}^1_{31}\}\Re\{t\}) + \Im\{\hat{\Omega}^1_{31}\}(-1 + \Im\{t\}^2 - \Re\{t\}^2))s^2 +$$
$$2(\Re\{t\} * (\hat{\Omega}^0_{31} + 2\Im\{\hat{\Omega}^1_{31}\}\Im\{t\}) + \Re\{\hat{\Omega}^1_{31}\}(1 + \Im\{t\}^2 - \Re\{t\}^2))s + \tag{12}$$
$$2\Re\{\hat{\Omega}^1_{31}\}\Im\{t\}\Re\{t\} - \hat{\Omega}^0_{31}\Im\{t\} + \Im\{\hat{\Omega}^1_{31}\}(1 - \Im\{t\}^2 + \Re\{t\}^2) = 0$$

Assuming $\Re\{b\}>0$ without loss of generality, solving Eqs 11 and 12, and using identity 6 provides eight pairs of values for $a$ and $b$:

$$b = \frac{1 + is}{\sqrt{(1 + |t|^2)(1 + s^2)}} \tag{13}$$

$$a = \frac{1 - is}{\sqrt{(1 + |t|^2)(1 + s^2)}} \cdot t \tag{14}$$

which correspond to eight rotations defined by matrix 5. (N.B. The form of Eqs 11 and 12 depends only on the indices $l$ and $m$, and therefore can be derived analytically in advance for efficient computation at run-time.)

Finally, the rotated 3D Zernike moments can be obtained from:

$$(\Omega_R)_{nl}^m = c_l^m (aa^*)^l \left(\frac{a}{b}\right)^m \sum_{\mu=-l}^{l} \hat{\Omega}_{nl}^\mu \left(-\frac{a}{b^*}\right)^\mu \sum_k \binom{l-\mu}{k-\mu}\binom{l+\mu}{k-m}\left(-\frac{bb^*}{aa^*}\right)^k \tag{15}$$

## BioZernike descriptors

BioZernike descriptors include two rotation-invariant shape descriptors: one based on the Canterakis Norms (CNs) and one based on the simple geometric features (GEO). In addition, we provide a CN-based alignment descriptor (Fig 3).

For the 3D Zernike moments calculation, the structure coordinates are converted to the volumetric representation as follows. First, the grid width is chosen in the range 0.25Å–16Å to keep the volume's average dimension between 50Å and 200Å, if possible. Subsequently, for every representative atom a Gaussian density is placed into the volume that corresponds to the amino acid/nucleotide weight and spherically averaged size. Representative atoms are defined as C$\alpha$ for amino acids and backbone phosphate groups for nucleotides. The volume is scaled into a unit sphere centered at the volume's center of mass with the scaling coefficient defined as 1.8 times the structure's gyration radius. Zernike moments are calculated up to the order of 20. CNs of orders $n = 2, 3, 4, 5$ are computed by setting $(\hat{\Omega}_R)_{n2}^2 = 0$ if $n$ is even and $(\hat{\Omega}_R)_{n3}^3 = 0$ if $n$ is odd. As the absolute values of the multiple solutions are averaged in each case, the third degree of freedom is lost and choice of a particular $\Im\{\hat{\Omega}_{nl}^m\} = 0$ has no effect. Every such CN for order $N = 20$ yields a vector of size 946 (expansion of the 3 indices $n$, $l$, $m$ with negative $m$ indices omitted), as opposed to 121 parameters obtained for a 3DZD (where index $m$ is not present). The final CN-based descriptor is a concatenation of the CNs of chosen orders and has 3784 components.

For the vector of geometric features GEO, we calculate the distance distribution from the center of mass of the structure to all its representative atoms. Next, we include in the vector moments of this distribution: standard deviation, skewness, kurtosis, as well as $10^{th}$, $20^{th}$, ...$90^{th}$ percentiles. In addition, we include the structure radius of gyration, nominal molecular weight, and standard deviation of the coordinates along the principal axes, corresponding to the dimensions of the structure. The final GEO descriptor has 17 components.

The alignment descriptor consists of two components: complete 3D Zernike moments calculated up to the order of 6 and the coordinates of the structure's center of mass (required because this information is not preserved by the volume scaling procedure). To perform structure alignment, we compute all possible CNs of the given moments and find a normalization

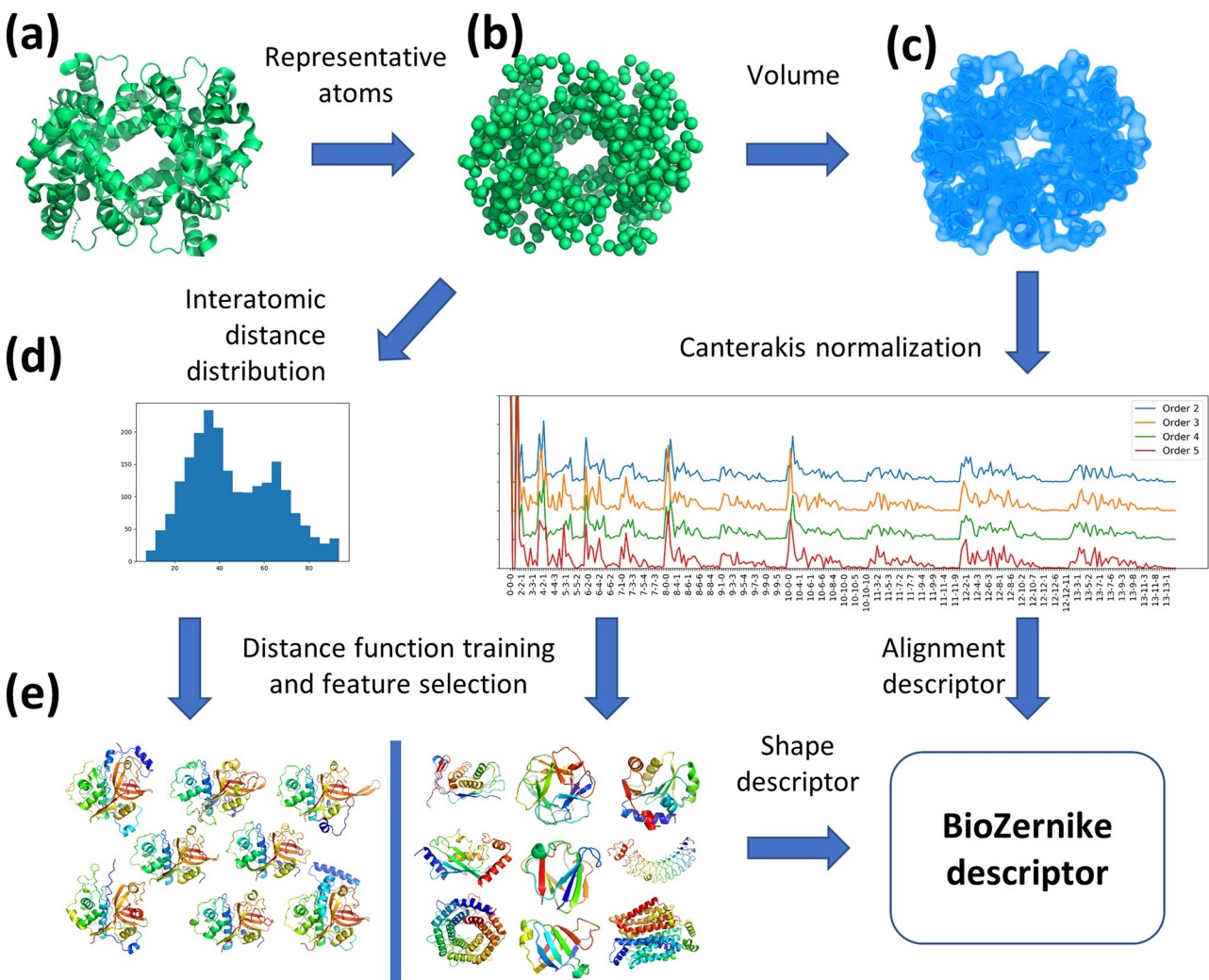

**Fig 3. BioZernike descriptors workflow.** Every atomic structure in the PDB (a) is converted to a volume by selecting representative atoms per residue (b) and placing a gaussian density in their place (c). The geometric features (GEO) can be calculated directly from the representative atoms coordinates, whilst the Zernike moments and their Canterakis Norms of various orders are calculated out of the volume (note that different normalizations are offset in $y$ axis for clarity) (d). The vector of concatenated geometric features and CNs of selected orders constitute the composite BioZernike shape descriptor. The distance between descriptors (composing both GEO and CNs) is calculated by learning optimal weights on a training set (e). The alignment descriptor is obtained directly from the CNs.

$R_{opt}$ (and the induced rotation) that minimizes distance $D(s_1, s_2|R)$ between the rotated moments of the structures $s_1$ and $s_2$ as follows:

$$D(s_1, s_2|R) = \sum_{nlm} \frac{|(\Omega_R)_{nl}^m(s_1) - (\Omega_R)_{nl}^m(s_2)|}{|(\Omega_R)_{nl}^m(s_1)| + |(\Omega_R)_{nl}^m(s_2)| + 1} \tag{16}$$

Then the optimal rotation $R_{opt}$ is selected by

$$R_{opt}(s_1, s_2) = \underset{R \in CN}{\arg\min}\, D(s_1, s_2|R) \tag{17}$$

The Eqs 16 and 17 generalize trivially to an arbitrary number of structures.

After applying the rotation, the structures are superposed using coordinates of their centers of mass.

## Shape descriptors distance function

For the distance function training, we prepared a CATH-based dataset as follows: A non-redundant subset of CATH domains with up to 40% sequence identity was obtained from www.cathdb.info. Homologous super-families with fewer than 20 members were removed. Structures within each superfamily were superposed with TM-align using an all-*versus*-all strategy. A representative domain was selected for each superfamily with the maximal median TM-score of alignments to other members of the superfamily. The dataset was subsequently pruned so that the domains within each superfamily align with the representative domain with TM-score at least 0.75. Finally, we selected super-families starting from the most populous ones, which have TM-score between their representative domains at most 0.45. This procedure yielded 2685 structures (divided among 151 families). All-*versus*-all comparisons produced a training set with 46572 'positive' (same super-family) and 3556698 'negative' (different super-families) data points.

We defined the distance function for the composite BioZernike feature vectors as:

$$D(\mathbf{g_1}, \mathbf{m_1}, \mathbf{g_2}, \mathbf{m_2}) = \sum_{i=1}^{N_g} w_g(i) \frac{2|g_1(i) - g_2(i)|}{1 + |g_1(i)| + |g_2(i)|} + \sum_{i=1}^{N_m} w_m(i)|m_1(i) - m_2(i)|$$

where $\mathbf{g_1}$ and $\mathbf{g_2}$ are the geometric feature vectors being compared, $\mathbf{m_1}$ and $\mathbf{m_2}$ are the CN-based feature vectors, and $\mathbf{w_g}$ and $\mathbf{w_m}$ are the respective weights.

The weights were fitted to the training set using regularized logistic regression. 10-fold cross-validation was performed on the superfamily level. The regularization parameter that maximized the Matthews correlation coefficient of the predictions of the excluded data was used for the final training with the entire dataset. Learned weight coefficients were constrained to non-negative values, which led to sparse solutions (*e.g.*, 1458 weights were non-zero after the final training on the CATH dataset).

Importantly, the obtained distance function is by no means definitive, but rather an illustration of a general approach. The procedure can (and should) be repeated with the problem-specific training sets, yielding appropriate functions based on the BioZernike descriptors.

## Benchmarking

**Domains.**   The domain test set was prepared based on the independent ECOD subset using the same procedure as for the CATH-based training set. Additionally, if an F-group representative domain could be aligned to any CATH superfamily representative domain with TM-score 0.75 or more, the group was excluded from the test set. Ultimately, 761 domain structures (divided among 34 families) remained. All-*versus*-all comparisons resulted in 13603 'positive' and 275577 'negative' data points.

**Assemblies.**   500 biological assemblies were randomly selected from all PDB entries such that no two assemblies have density correlation score [34] larger than 0.5, to ensure distinct shapes. Afterwards, normal mode analysis from the ProDy package [35] was used to sample 4 additional conformations of each assembly, resulting in 2500 total structures evenly split into 500 classes. As in the domain set evaluation, all-*versus*-all comparisons were assessed, yielding 5000 'positives' and 3118750 'negatives'.

**Reference methods.**   3D-Surfer 3DZD descriptors were obtained directly from the web server using default parameters. During the course of this work, we discovered a bug in the original 3DZD library [32], which caused the invariants of the same order to be cumulative. For the sake of fairness, we corrected the descriptors obtained from the 3D-Surfer server for this bug and notified the server maintainers (the problem has since been solved). The

correction somewhat improved the discrimination power (S1 Fig). Euclidean distance was used as the scoring function.

As Omokage score is not available for use with arbitrary structures, we implemented the procedure described in [23]. First, all structures were converted to representative point-sets with Situs v3.1 [36]. Second, we calculated the iDR profiles and implemented the Omokage score according to [23]. The implementation was independently validated using 1000 random comparisons between structures selected from PDB30. Comparisons were scored with our implementation and the PDBj omokage-pairwise web service. As shown in S2 Fig, scoring is consistent between the implementations and insignificant deviations are likely due to the different versions of the Situs package used.

## Results

### Protein superposition and similarity assessment with complete 3D Zernike moment invariants

We achieve fast similarity computation using protein structural descriptors. Robust descriptors must capture information relevant to their intended use (e.g., binding sites for virtual drug screening, solvent-accessible surfaces for protein docking, structural organization for establishing functional or evolutionary relationships) while being inexpensive to compute, quick to compare with other descriptors, and readily interpretable. Most high-throughput structure analysis pipelines involve balancing the tradeoff between speed and accuracy of the underlying representation.

3D Zernike moments, derived by Canterakis in [31], allow decomposition of an arbitrary volumetric function $f(x)$ into a set of parameters $\Omega_{nl}^{m}$ (see Methods). These parameters are independent, insensitive to noise, and, importantly, embody a hierarchy of shape representation. The latter property is of particular significance, as it enables intuitive interpretation of the information content in the moments of certain order (Fig 1).

Limiting their use, 3D Zernike moments are not invariant under rotation. While special properties of the spherical harmonic functions can be exploited to align two sets of moments, the resulting procedure is slower than classical coordinate-based methods [30]. A popular software library [32] implemented the 'trivial' rotation invariant descriptors from 3D Zernike moments, $i.e.$, norms of the vectors $\mathbf{\Omega}_{nl} = (\Omega_{nl}^{l}, \Omega_{nl}^{l-1}, ..., \Omega_{nl}^{-l})$. We will refer to these descriptors as 3DZDs (3-Dimensional Zernike Descriptors) for consistency with prior work [27]. While this approach is straightforward and has proven to be widely applicable [26, 37], the information loss is obvious: every $(2l + 1)$-dimensional vector of parameters is reduced to a single invariant. These simpler 3DZD invariants are the base of 3D-surfer [26], the first widely available tool that made use of the Zernike moment decomposition for protein shape matching.

In his work Canterakis [33, 31] did derive a special normalization of the 3D Zernike moments, making them rotationally invariant. However, we found that this normalization does not perform well for the shapes of proteins and macromolecular complexes due to the abundance of symmetric oligomeric arrangements.

Here we generalize the approach of Canterakis by developing normalization routines with wider applicability. These routines yield complete, rotationally invariant 3D Zernike moments (referred to hereafter as Canterakis Norms or CNs). Conceptually, a CN rotates an object so that selected moments become equal to predefined values. This orients an object in a uniquely determined standard position (Fig 2a).

CNs immediately give rise to a computationally inexpensive global structure alignment. Indeed, if the same moments are normalized to their standard values for two objects, their induced standard positions are likewise equivalent (Fig 2a). By normalizing moments from

various orders, we obtain alternative alignments that may be more or less suitable for a particular application (Fig 2b). Moreover, since the alignment is performed not between any two objects, but from every object to its standard position, an arbitrary number of structures may be aligned in linear time (Fig 2c).

It has to be noted that the system of polynomial equations mentioned above has multiple solutions. While there is a theoretically sound approach to break this ambiguity described in [33] (fixing signs of the selected moments), in practice it is not universally applicable because we calculate an approximation of the Zernike functions on a discrete volumetric grid. Thus, if the selected moment's value is small at the outset, the gridding effect may lead to its sign changing randomly based on the initial orientation. Nevertheless, we found that for structure alignment, we can either choose the ambiguity-breaking rule at runtime, based on the observed moment magnitudes, or circumvent the problem entirely by testing all possibilities with negligible loss of performance.

Finally, but most importantly, the complete moments of objects oriented in the same standard position are *comparable*. This critical property underpins our novel search procedure presented below.

## Search procedure and evaluation

Our search procedure is depicted schematically in Fig 3, and relies on the newly designed Bio-Zernike descriptor, a composite descriptor based on several CNs augmented with geometric features.

The simpler 3DZD descriptors are usually calculated for an object surface, following the original implementation [32]. We reasoned that the density distribution contains valuable information for protein structure similarity retrieval. Therefore, we use simulated volumes (Fig 3c) as the basis for 3D Zernike moments calculation. Thus our newly introduced system has two important differences with 3D-Surfer [26]: use of full volumetric data instead of surface only and use of the complete CN invariants instead of the simpler 3DZD invariants.

CNs of different orders may be more or less appropriate for various shapes. Moreover, resolving ambiguity for multiple solutions depends on a particular symmetry that an object may possess. To make the CN-based descriptor versatile while retaining performance, we used several CNs of orders 2 to 5 and then average the absolute values of the solutions (Fig 3d).

The 3D Zernike moments are defined for objects scaled to a unit ball which loses size-related information. To compensate for this fact, we developed a geometry-based descriptor (GEO). It includes features that can be quickly obtained from the set of representative atoms, such as structure dimensions along its principal axes or statistical properties of the interatomic distance distribution (see Methods).

Together the CN-based and the GEO descriptors constitute what we term a BioZernike descriptor. In order to judge similarity of 2 structures, we developed a distance measure that compares their BioZernike descriptors. We hypothesized that the often-used Euclidean distance is suboptimal choice for comparing 3D Zernike moments-based descriptors, because of the hierarchical structure of the representation (Fig 1). Instead, we have followed a machine learning approach to determine weights for the descriptor components using a training set. We used a non-redundant subset of CATH [4] families for this purpose (see Methods).

Retrieval of similar structures was evaluated on a non-redundant subset of ECOD [5] and on a set of biological assemblies with distinct density shapes. 3D-Surfer [26] and Omokage [23] were selected for benchmarking purposes, as both operate on similar principles and represent the current state of the art. 3D-Surfer uses 3DZD descriptors and Euclidean distance to compare them, considering only the solvent-accessible surface of a protein. Omokage scoring

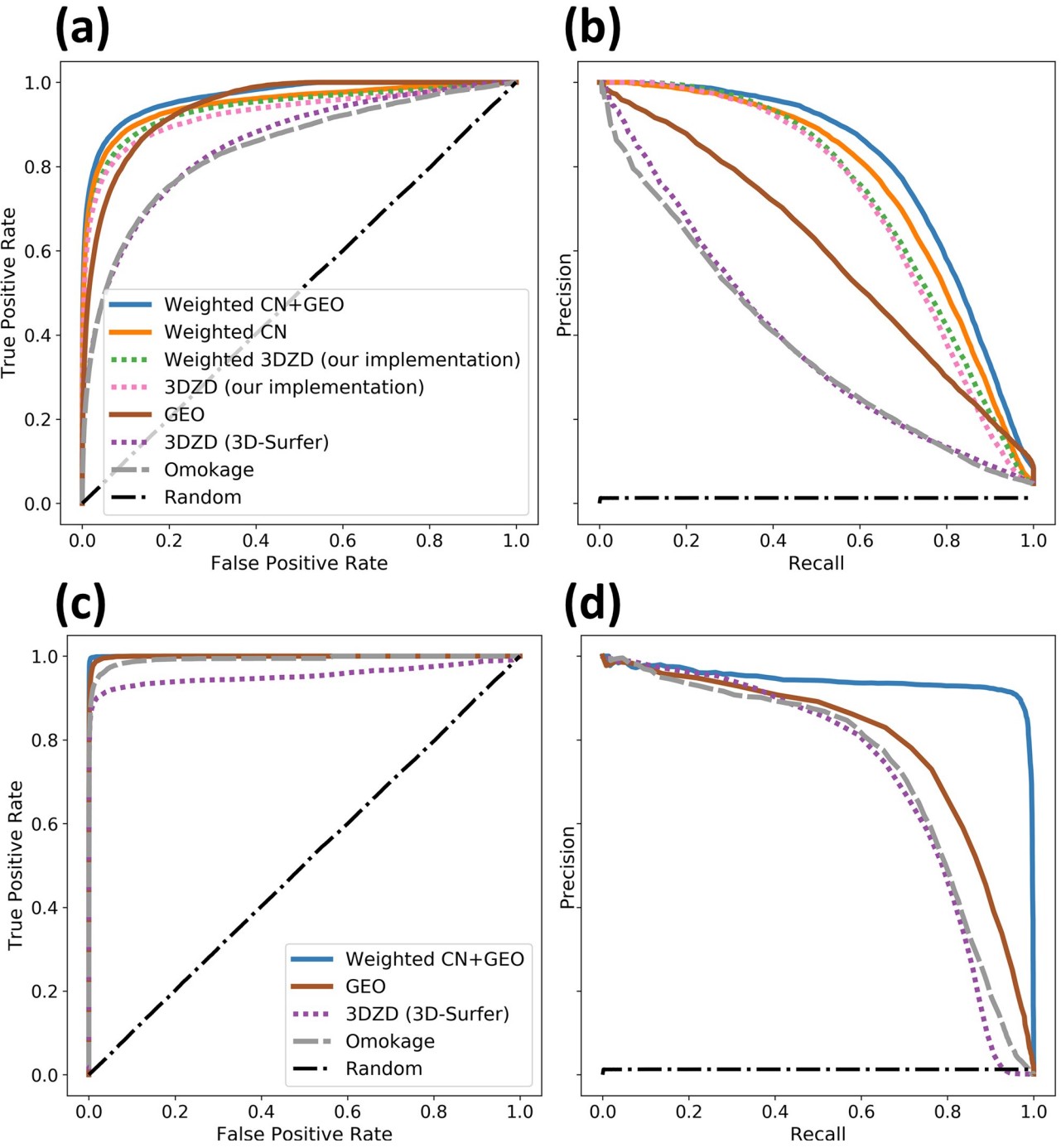

**Fig 4. Shape retrieval performance comparison of the BioZernike descriptors to the 3D-Surfer and Omokage descriptors.** Non-redundant sets of domains (a,b) and assemblies (c,d) were used for the evaluation. Receiver operating characteristic (a,c) and precision-recall (b,d) curves are shown. For the domain set, performance of the Canterakis norms (CNs) is plotted separately as well as in conjunction with the geometry descriptor (GEO). '3DZD (our implementation)' corresponds to our implementation of the 3DZD descriptors that takes into account the whole density distribution, rather than the protein surface only.

utilizes properties of the interatomic distance distribution function (similar to our GEO score).

As shown in Fig 4, 3DZD descriptors applied directly to the density volume retain significantly more information pertaining to similarities and differences within the protein domain

**Table 1. Area under curve (AUC) for receiver operating characteristic (ROC) and precision-recall (PR) curves, and the maximal achievable Matthews correlation coefficient (MCC) are reported for each evaluation plotted in Fig 4.**

| | BioZernike components | | | | | Reference methods | |
|---|---|---|---|---|---|---|---|
| | Weighted CN | Weighted 3DZD | Density 3DZD | GEO | Weighted CN+GEO | 3DSurfer 3DZD | Omokage |
| **Domains** | | | | | | | |
| ROC AUC | 0.95 | 0.94 | 0.93 | 0.94 | 0.97 | 0.85 | 0.84 |
| PR AUC | 0.76 | 0.73 | 0.71 | 0.59 | 0.79 | 0.39 | 0.38 |
| MCC | 0.69 | 0.66 | 0.65 | 0.54 | 0.72 | 0.38 | 0.38 |
| **Assemblies** | | | | | | | |
| ROC AUC | – | – | – | 1.0 | 1.0 | 0.95 | 0.99 |
| PR AUC | – | – | – | 0.79 | 0.94 | 0.71 | 0.71 |
| MCC | – | – | – | 0.75 | 0.93 | 0.70 | 0.71 |

families. Retrieval performance is further improved by 1) using the custom distance function obtained via the training set, 2) using CNs instead of 3DZDs, and 3) augmenting CNs with GEO scoring. (AUCs for all evaluated methods are listed in Table 1).

## BioZernike library

An important result of this study is an open-source, customizable library that implements all routines required to obtain a BioZernike descriptor starting from a protein structure https://github.com/biocryst/biozernike. It is written in Java language and can be integrated into any project written in a JVM-compatible language. For instance it can be used together with Bio-Java to take advantage of its comprehensive structural bioinformatics capabilities [19].

The BioZernike library includes structure-to-volume conversion based on the `gmconvert` program [34]. We implemented dynamic scaling of the volume grid size to make both speed and precision equally suitable for smaller proteins and larger macromolecular assemblies. 3D Zernike moments were implemented after [32], with further optimization for batch processing. In addition to the classical 3DZD invariants, the library contains routines for calculating CN-norms introduced here and their application to multiple structure alignments.

The library is developed and continuously validated for processing of large amounts of structural data, such as those at RCSB PDB, which leads to a highly optimized and efficient implementation. For example, calculating a full BioZernike descriptor for PDB ID 5J7V (the largest macromolecular assembly represented in the PDB archive at the time of writing; 8280 component homo-oligomer containing 5,340,600 amino acid residues) takes $\sim 10$ seconds. For a more typical oligomeric PDB structure, such as PDB ID 4HHB (hemoglobin $\alpha_2 \beta_2$ hetero-tetramer containing 574 residues) the processing time is $\sim 30$ milliseconds. The full processing for the entire archive as of November 2019 (all assemblies and all polymeric chains) takes 7 hours using 6 parallel threads. The time needed for descriptors comparison and moment alignment is negligible. The performance of the BioZernike library is showcased on the website shape.rcsb.org. The library's comprehensive implementation, together with its flexibility and speed makes it especially useful for developers who wish to create novel applications for classification and comparison of structural data.

The rcsb.org main website also makes use of the BioZernike library since the April 2020 release. The integration enhances the applicability of this system by combining structure search with other types of searches. Two structure search modes are made available at rcsb.org: "strict" and "relaxed". The modes correspond to two different threshold sets, based on training against the assemblies dataset (see Methods). "Strict" corresponds to maximal Matthews Correlation Coefficient (about 98% recall), whilst "relaxed" corresponds to 99.9% recall.

### Case studies

**Similar quaternary structure but different subunits.** A very interesting set of comparisons is that of folds that are conserved globally across evolution while the subcomponents have been re-arranged in different ways as a result of gene fusion or duplication. The different biological assemblies are thus very similar overall but have different stoichiometries.

A striking example of this property is the Macrophage migration inhibitory factor (MIF). These tautomerase enzymes are conserved across the entire tree of life (Fig 5a). The assembly fold is composed of $\beta - \alpha - \beta$ motifs, with a central $\beta$ barrel formed by the different subunits coming together around a 3-fold axis of symmetry. Overall the assembly has quasi D3 point group symmetry. In eukaryotes (human: 4GRO; mouse: 1MFI; nematode: 2OS5) and cyanobacteria (2XCZ *Prochlorococcus marinus*) the enzyme is trimeric with a pseudo 2-fold symmetry within the subunits. In some bacteria the enzyme is hexameric, with both A6 (2X4K *Staphylococcus aureus*) and A3B3 (3EJ9 *Pseudomonas pavonaceae*) systems known.

Another well-known case in this category is that of the DNA-clamps (overall quasi D6 symmetry, with A3 stoichiometry in archaea and eukaryotes and A2 in bacteria) [38], also detectable with our system.

Domain swaps are a further example that belong to this category and represent a widespread phenomenon in structural biology [39]. These assemblies do not present different stoichiometries but rather conformational changes in the subunits that allow for domains to be swapped with partner subunits. Conservation of quaternary structures between swapped and non-swapped assemblies is also detectable with our BioZernike system.

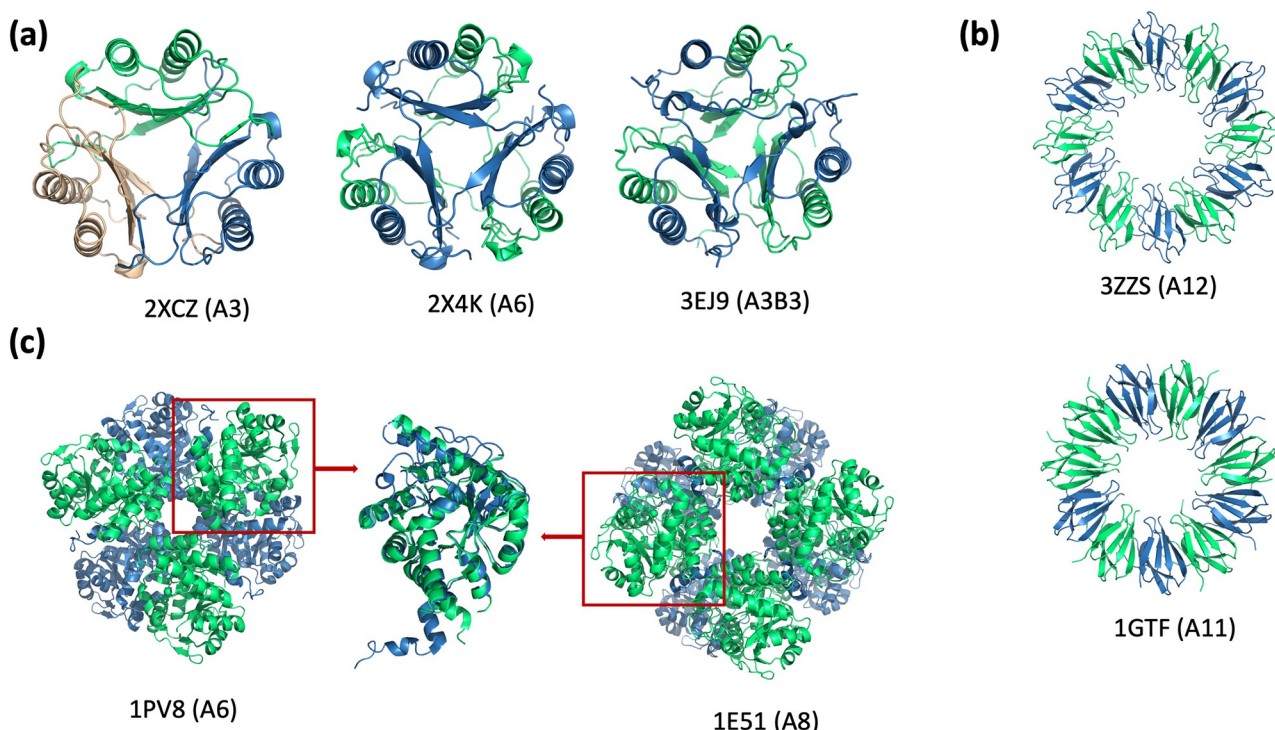

**Fig 5. Structure search modes.** (a,b) Global shape search finds similar assemblies regardless of a particular stoichiometry. (c) Search-by-parts mode allows discovery of the 'transformer' proteins that form different assemblies from similar components.

**Different quaternary structure but similar subunits.** Another case we can easily study and find using the BioZernike system involves divergent quaternary structure assemblies composed by the identical or highly similar subunits.

For example, the TRAP (trp RNA-binding attenuation protein) proteins from *Geobacillus stearothermophilus* have been determined to have both A11 and A12 stoichiometries, respectively with C11 and C12 point group symmetries (Fig 5b). Since both the global shape (low order Zernike moments) and the details (high order Zernike moments) are matching well, this case does not represent a problem for our retrieval system.

Other well-known cases were revealed through a mutation or an environmental change (*e.g.*, pH) that favored one assembly relative to the other. One case in which conformational changes in the individual subunits lead to very different quaternary structures is porphobilinogen synthase (PBGS) [40], which exists as both a hexamer (1PV8) and octamer (1E51) (Fig 5c). Here, a single point mutation F12L dramatically shifts the equilibrium from octamer to hexamer, which produces an altered pH-rate profile; the hexameric assembly is active only at basic pH. Proteins, like PBGS, that can come apart, change subunit conformation, and reassemble differently with functional consequences have been named morpheeins, and can reflect a dissociative allosteric mechanism [41].

Our search system makes possible automatic identification of such proteins. In this case, the search can be performed only on parts of an assembly, such as an individual chain or a domain. Moreover, a distance function can be designed specifically to focus on higher-order Zernike moments so that structural features of the sub-components are weighted more heavily than the overall shape of the assembly.

## Discussion

Protein structure retrieval and alignment with volumetric data provides several advantages *versus* traditional atomic-data based systems. First, it avoids the challenge of dealing with chain topology (arrangement of secondary structure components along the polypeptide chain) important in many biological systems (*e.g.*, in circular permutants). Second, it obviates the problem of quaternary structure topology, be they local changes as in domain swapping or global changes with subunits that merge or split while conserving the overall quaternary arrangement. A further advantage of our method is that it automatically solves the chain matching problem [21]. These advantages combined with speed, yield a system that compares favorably to quaternary structure search and alignment tools in terms of scalability [14, 42, 20, 19]. Moreover, this method does not rely on atomic models and thus can be applied directly to the growing number of experimental maps obtained using 3D electron microscopy (3DEM) and available from the EMDB data resource (www.ebi.ac.uk/pdbe/emdb/). In fact the properties of 3D Zernike moments make them very suitable for experimental data: robustness to noise and hierarchical description from low to high resolution features. On the other hand, 3DEM maps present a few issues to address in a practical application, such as selection of an appropriate map contour level and consistent radial scaling of the volume into the unit sphere, which are complicated by a relatively high degree of noise density in a typical EM volume.

At the same time, volumetric data preserves information content far better than shape based on surface representation. Thus, as the benchmarks above demonstrate, there are clear advantages of our volume-based system when compared to surface based methods [26, 27]. Trivial examples such as hollow viral capsid versus a full spherical protein assembly can readily exemplify the difference.

On the negative side, one important disadvantage of this method is the inability to find local matches, e.g. finding a conserved domain between 2 chains that have different domain

architecture and thus do not match globally. However there are ways of working around the problem, for instance domain decomposition of both query and database to perform searches specific to domains.

A key to the success of our method is the fact that the newly derived Canterakis normalizations (CNs) preserve much more information than the widely used 3DZDs. This is clearly demonstrated by the alignments that are naturally obtained from CNs.

As shown in the benchmark, our system outperforms other fast descriptor-based search approaches in terms of both precision and sensitivity. At the same time, it allows real-time (milliseconds) PDB archive-wide retrieval without the need to resort to ad-hoc strategies for speeding up the calculation. Run times of currently available services (Dali, TopSearch, PDBe-Fold) are measured in seconds or minutes solely because of additional speed-up strategies like pre-clustering or parallelization. Our system's faster performance applies equally to user input atomic coordinates, adding only minimal overhead typically measured in milliseconds. Such a system constitutes a valuable tool for structural biologists, allowing for real-time hypotheses generation at the conclusion of a structure determination campaign.

Importantly, the speed and accuracy of this method opens up the possibility of automated structural classification at any level, an avenue that we shall explore in future work. One interesting application is multiple structure profiles: using normalized complete moments to parameterize entire protein families. A related 'consensus shapes' notion was introduced in [30], however we feel that using descriptor profiles rather than simple averages is the key. Akin to multiple sequence profiles, it would enable more sensitive searching through profile-to-profile alignments. Moreover, shape variation can be easily studied and visualized for an ensemble of structures. Further applications include scalable multiple structure alignment, alignment of 3DEM maps, and automated structural model-to-electron density (or 3DEM) map fitting.

## Supporting information

**S1 Fig.** (a) The 3DZD library bug results in characteristic non-decreasing wave pattern (red) of the descriptor, often found in literature. The same descriptor without the bug is shown for comparison in blue. (b) Precision-recall curve using Euclidean distance on the test set improves with the corrected version of the descriptor.
(TIF)

**S2 Fig. Omokage score validation.** The score calculated by our implementation (axis $X$) is plotted versus the score obtained from the PDBj server (axis $Y$) for 1000 random comparisons between structures in PDB.
(TIF)

## Author Contributions

**Conceptualization:** Dmytro Guzenko, Jose M. Duarte.

**Formal analysis:** Dmytro Guzenko.

**Funding acquisition:** Stephen K. Burley.

**Investigation:** Dmytro Guzenko, Jose M. Duarte.

**Methodology:** Dmytro Guzenko.

**Project administration:** Stephen K. Burley, Jose M. Duarte.

**Resources:** Stephen K. Burley, Jose M. Duarte.

**Software:** Dmytro Guzenko, Jose M. Duarte.

**Supervision:** Stephen K. Burley, Jose M. Duarte.

**Validation:** Dmytro Guzenko.

**Writing – original draft:** Dmytro Guzenko, Jose M. Duarte.

**Writing – review & editing:** Dmytro Guzenko, Stephen K. Burley, Jose M. Duarte.

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
