## [Decision Letter · Decision Letter 0]

27 Feb 2020

Dear Dr Duarte,

Thank you very much for submitting your manuscript "Preparing for the 3D data deluge: real time structural search at the scale of the PDB and beyond" for consideration at PLOS Computational Biology. As with all papers reviewed by the journal, your manuscript was reviewed by members of the editorial board and by several independent reviewers. The reviewers appreciated the attention to an important topic. Based on the reviews, we are likely to accept this manuscript for publication, providing that you modify the manuscript according to the review recommendations.

Sincerely,

Charlotte M Deane

Associate Editor

PLOS Computational Biology

Arne Elofsson

Deputy Editor

PLOS Computational Biology

[LINK]

Reviewer's Responses to Questions

**Comments to the Authors:**

Reviewer #1: Review comments attached as pdf. Predominantly, what is needed is the clarification of the method through the use of non-ambiguous language.

Reviewer #2: The authors present an approach for fast retrieval of similar structures from PDB, using 3D Zernike moments. The proposed approach is an improvement over existing tools and the speed of the method is a big advantage and shows better performance in authors benchmark. The implementation seems robust and is a useful tool for the community. I have a few minor points that will help with user interpretation of results.

1) How does the approach perform (and rank) in case of partial alignments where only part of the larger protein is matched?

2) What are the safe thresholds of the total score to find close and distantly related structures? How much does Zernike descriptors contribute to the total score in general? A discussion would help users.

3) The method can deal with EM volumes but the current implementation and tests doesn’t include these. It is useful to discuss this although I noted that this is mentioned as part of future developments.

**Have all data underlying the figures and results presented in the manuscript been provided?**

Reviewer #1: No: The approach used for generating the dataset is included but seemingly not the list of structures that resulted. list/spreadsheet of optimised zernike/geometric distance metric weights could also be included.

Reviewer #2: No: The benchmark datasets used for tests

PLOS authors have the option to publish the peer review history of their article (what does this mean?). If published, this will include your full peer review and any attached files.

Reviewer #1: No

Reviewer #2: No
---

## [Editor Report · Decision Letter 1]

20 May 2020

Dear Dr Duarte,

We are pleased to inform you that your manuscript 'Real time structural search of the Protein Data Bank' has been provisionally accepted for publication in PLOS Computational Biology.

Best regards,

Charlotte M Deane

Associate Editor

PLOS Computational Biology

Arne Elofsson

Deputy Editor

PLOS Computational Biology

---

## [Editor Report · Acceptance letter]

30 Jun 2020

PCOMPBIOL-D-19-01993R1 

Real time structural search of the Protein Data Bank

Dear Dr Duarte,

I am pleased to inform you that your manuscript has been formally accepted for publication in PLOS Computational Biology. Your manuscript is now with our production department and you will be notified of the publication date in due course.

With kind regards,

Laura Mallard
